# Acral Melanoma Is Infiltrated with cDC1s and Functional Exhausted CD8 T Cells Similar to the Cutaneous Melanoma of Sun-Exposed Skin

**DOI:** 10.3390/ijms24054786

**Published:** 2023-03-01

**Authors:** Saraí G. De Leon-Rodríguez, Cristina Aguilar-Flores, Julián A. Gajón, Alejandra Mantilla, Raquel Gerson-Cwilich, José Fabián Martínez-Herrera, Benigno E. Rodríguez-Soto, Claudia T. Gutiérrez-Quiroz, Vadim Pérez-Koldenkova, Samira Muñoz-Cruz, Laura C. Bonifaz, Ezequiel M. Fuentes-Pananá

**Affiliations:** 1UMAE Hospital de Especialidades, Centro Médico Nacional Siglo XXI, Instituto Mexicano del Seguro Social, Unidad de Investigación Médica en Inmunoquímica, Mexico City 06720, Mexico; 2Posgrado en Ciencias Biológicas, Universidad Nacional Autónoma de México, Mexico City 04510, Mexico; 3UMAE Hospital de Pediatría, Centro Médico Nacional Siglo XXI, Instituto Mexicano del Seguro Social, Mexico City 06720, Mexico; 4Posgrado en Ciencias Bioquímicas, Universidad Nacional Autónoma de México, Mexico City 04510, Mexico; 5Servicio de Patología, Hospital de Oncología Centro Médico Nacional Siglo XXI, Instituto Mexicano del Seguro Social, Mexico City 06720, Mexico; 6Cancer Center, Medical Center American British Cowdray, Mexico City 01120, Mexico; 7Latin American Network for Cancer Research (LAN-CANCER), Lima 11702, Peru; 8International Cancer Center, Campus Santa Fe, Mexico City 05348, Mexico; 9UMAE Hospital de Especialidades, Centro Médico Nacional General Manuel Avila Camacho, Puebla 72000, Mexico; 10Laboratorio Nacional de Microscopía Avanzada-IMSS, División de Desarrollo de la Investigación, Centro Médico Nacional Siglo XXI, Instituto Mexicano del Seguro Social, Mexico City 06720, Mexico; 11Coordinación de Investigación en Salud, Centro Médico Nacional Siglo XXI, Instituto Mexicano del Seguro Social, Mexico City 06720, Mexico; 12Unidad de Investigación en Virología y Cáncer, Hospital Infantil de México Federico Gómez, Mexico City 06720, Mexico

**Keywords:** melanoma, acral, CD8, cDC1s, PD-1, PD-L1, exhaustion

## Abstract

Acral melanoma (AM) is the most common melanoma in non-Caucasian populations, yet it remains largely understudied. As AM lacks the UV-radiation mutational signatures that characterize other cutaneous melanomas, it is considered devoid of immunogenicity and is rarely included in clinical trials assessing novel immunotherapeutic regimes aiming to recover the antitumor function of immune cells. We studied a Mexican cohort of melanoma patients from the Mexican Institute of Social Security (IMSS) (n = 38) and found an overrepresentation of AM (73.9%). We developed a multiparametric immunofluorescence technique coupled with a machine learning image analysis to evaluate the presence of conventional type 1 dendritic cells (cDC1) and CD8 T cells in the stroma of melanoma, two of the most relevant immune cell types for antitumor responses. We observed that both cell types infiltrate AM at similar and even higher levels than other cutaneous melanomas. Both melanoma types harbored programmed cell death protein 1 (PD-1^+^) CD8 T cells and PD-1 ligand (PD-L1^+^) cDC1s. Despite this, CD8 T cells appeared to preserve their effector function and expanding capacity as they expressed interferon-γ (IFN-γ) and KI-67. The density of cDC1s and CD8 T cells significantly decreased in advanced stage III and IV melanomas, supporting these cells’ capacity to control tumor progression. These data also argue that AM could respond to anti-PD-1-PD-L1 immunotherapy.

## 1. Introduction

Cutaneous melanoma is a type of skin cancer that arises from melanocytes, the specialized cells producing the pigment melanin. Melanoma is the deadliest form of skin cancer, accounting for 60% of skin cancer deaths worldwide and 80% in Mexico [1,2]. Melanoma is a neoplasm with alarming invasive potential. Although most of the early stages of this malignancy have a good prognosis (stages I or II), most cases are usually diagnosed at late stages, particularly in countries with limited resources. Conventional therapies, namely surgical excision and chemotherapy, fail to offer long-term benefits to patients with advanced melanoma [3]. 

Melanoma usually develops in sun-exposed areas of the skin as a consequence of ultraviolet (UV) radiation exposure, disproportionately affecting fair-skinned individuals [4]. For instance, Caucasians from Australia and New Zealand, where UV radiation is higher due to ozone shield depletion, register the highest incidence of melanoma worldwide [4,5]. Cutaneous melanoma can also arise in areas of the skin not exposed to solar radiation, such as the palms, soles, and nails (acral melanoma, AM). These types of melanomas mainly affect dark-skinned populations [6,7,8,9]. While the incidence of cutaneous melanoma in sun-exposed areas (non-acral cutaneous melanoma, NACM) has remained stable for many years, the incidence of AM is increasing significantly, particularly in African, Asian, and Hispanic populations, traditionally considered at low risk [10,11]. Indeed, the Global Cancer Observatory (GCO) estimates that AM is the fastest-growing melanoma in Latin American countries [12]. 

The prevalence of NACM in the USA and Western Europe has increased awareness of this type of cancer. Due to its relationship with UV radiation, it is the most mutated type of cancer [13]. Due to this high rate of mutation, neoantigen expression drives strong antitumor immune responses [14,15,16]. Indeed, melanoma has helped to understand the mechanisms involved in tumor immunosurveillance, the particular roles of the innate and adaptive immune cells, and their interactions within the tumor microenvironment [17]. This is illustrated by the capacity of dendritic cells (DC) to present antigens, activate, and modulate T lymphocyte responses [18]. Particularly, conventional type 1 DCs (cDC1) are recognized as the most competent cells to activate CD8 T cells due to their enhanced ability to cross-present tumor antigens on MHC class I molecules [19,20,21,22]. CD8 T cells are the central antitumor immune cells because of their capacity to secrete cytotoxic molecules and effector cytokines (such as interferon-γ (IFN-γ)). The former induces the death of the target tumor cell, and the latter orchestrates optimal innate and adaptive immune responses [23]. Indeed, multiple studies support the notion that CD8 T cell scoring is central to inferring cancer prognosis [24,25,26,27].

Host immune responses are bound to fail during tumor evolution. Although there are many mechanisms of tumor immune-escape, arguably one of the most important is the exhaustion of the cytotoxic CD8 T cells [28,29]. These cells, when chronically exposed to antigenic stimuli, respond by gradually expressing increased levels of inhibitory checkpoint receptors (such as programmed cell death protein 1 (PD-1), T-cell immunoglobulin and mucin-domain containing-3 (TIM-3), CD39, and cytotoxic T-lymphocyte-associated protein 4 (CTLA-4), among many others) becoming progressively less functional and proliferative [30,31,32]. Furthermore, cDC1s express the PD-1 ligand (PD-L1), failing to activate T cells [33]. The importance of this phenomenon also lies in the new immune checkpoint blockade (ICB) therapies aiming to hinder PD-1-PD-L1 interactions, overcome exhaustion, and restore the functionality of CD8 T cells [34,35]. Due to its high neoantigen expression, NACM has been an ideal malignancy to test ICB approaches [36,37,38]. It has been documented that the density of cDC1s decreases while exhausted CD8 T cells increase as melanoma staging progresses [39]. Furthermore, PD-1-expressing CD8 T cells are among the main targets of classic immunotherapy with anti-PD-1 antibodies [28,40]. Thus, high infiltration of NACM with cDC1s and CD8 T cells and expression of PD-1 are markers of a better prognosis and a positive response to anti-PD-1 ICB, respectively [41,42,43,44,45].

Although different studies support an AM mortality rate higher than NACM, it is unclear whether it is intrinsically more aggressive and resistant to conventional chemotherapy or is more often diagnosed at later stages. Since most epidemiological, clinical, and molecular studies are based in high-income countries with a high incidence of NACM, we know significantly less about other types of melanoma, particularly the fast-growing AM. Furthermore, most of what we know about AM comes from cohorts with a minority of participants of non-Caucasian origin residing in high-income countries such as the USA, and there are only a handful of studies conducted with patients native to countries with a high incidence of this particular melanoma type. Due to its high incidence in developing countries, AM has limited access to novel therapeutic approaches such as ICB. As a result, there is little evidence of whether AM would also benefit from these approaches [46,47,48,49].

We recently reported the presence of infiltrating cDC1s in a Mexican cohort with a predominance of AM. The density of these cells was higher in patients controlling the disease than in those that progressed to metastatic stages, similar to NACM patients [50]. It has also been described that low numbers of tumor-infiltrating lymphocytes (TILs) were associated with poor prognosis in AM [51]. Furthermore, transcriptomic studies suggested that AM harbors lower levels of CD8 T cells than NACM, leaving the question of whether patients with this type of melanoma may also respond to immunotherapy [52]. In this study, we analyzed archival biopsy specimens of primary melanoma samples from a tertiary oncology hospital at the Mexican Institute of Social Security (IMSS) in Mexico City. We aimed to assess the frequency of AM and NACM and compare the level of infiltrating cDC1 and CD8 T cells and the expression of markers of exhaustion and function between both types of melanoma. We developed a multiparametric immunofluorescence (mIF) technique coupled with an artificial intelligence algorithm for image analysis. We found a disproportionately large presence of AM cases in our cohort, accounting for 74% of all samples. We did not observe differences in the absolute numbers of infiltrating cDC1s and CD8 T cells between AM and NACM. Instead, both types of melanoma differed in the level of infiltrate in early and advanced tumors, with a significant decrease in the infiltrate in stage III and IV melanomas. We also observed that the AM stroma and the tumor cells express PD-1 and PD-L1 exhaustion markers. Importantly, we observed two clusters of PD-1^+^ CD8 T cells, one highly proliferating and another producing IFN-γ. Altogether, these results support that AM is immunogenic and infiltrated with antitumor cDC1s and CD8 T cells at levels comparable to NACM.

## 2. Results

### 2.1. Study Cohort: Acral Melanoma Exceeded 3:1 Other Forms of Cutaneous Melanoma

We obtained archival paraffined melanoma blocks from tissue resection products collected during disease diagnosis. We impartially selected samples of AM and NACM from 2017 and 2018, collecting 38 samples that had sufficient tissue for analysis (Figure 1A). The classification of AM and NACM was made by the pathologist according to the primary lesion location. Even though we do not have information on the length of UV exposure, it is reported that the palms and soles are lowly exposed areas [53]. In Appendix A, we depict relevant demographic and clinical data for every patient in the cohort. Since we wanted to focus on cutaneous melanomas, we excluded mucosal and uveal melanomas from the study. We observed that AM represented 73.9% (n = 30) of all the cases, while NACM represented only 26.1% (n = 8) (Figure 1B). We noticed that AM tended to be diagnosed at advanced stages, with stages II (36.4%) and III (32.8%) representing most of the samples, whereas NACM was mainly diagnosed at stage I (66.6%) (Figure 1C). Delayed diagnosis due to the more occult location of AM represents an increased risk of an unfavorable outcome in our population. 

Since there is scarce literature on AM in our population, we compared this type of melanoma with NACM. We scanned hematoxylin and eosin (H&E)-stained tissue samples to evaluate their histological characteristics and to map areas with extensive infiltration of non-tumor cells. As AM is located in the nails, palms, and soles, we observe a thick epidermis and stratum corneum, as well as the absence of hair follicles. In contrast, NACM was characterized by areas with a thin epidermis and an abundance of follicles and sebaceous glands. In addition to these histological differences, both types of melanoma showed the presence of highly infiltrated areas spread throughout the tissue, particularly in areas close to the tumor (Figure 1D). These data confirm that AM is the most prevalent melanoma in the Mexican population and is usually diagnosed at advanced stages. Notably, the rich infiltrate of AM suggests that it also harbors immune cells. 

### 2.2. Acral Melanoma Is Infiltrated by Conventional Type 1 Dendritic Cells

We were surprised to find a robust infiltration in AM at similar levels to NACM. Due to its occult location and protection from the sunlight, some studies have supported a lower rate of mutation, hypothesizing less protection from immune cells compared with NACM [9,53]. We previously reported that cDC1s infiltrate the tumors of patients with metastatic melanoma (stage III or IV); however, we did not evaluate whether there was a difference in the density of these cells between AM and NACM [50]. Considering that only 26.1% of our study cohort were NACM cases and that only one patient was diagnosed at an advanced stage IV (Figure 1C), we decided to enrich this group by selecting five additional advanced NACM samples from patients from other IMSS and private hospitals. From now on, we will be using this enriched NACM cohort to assess and compare the density and phenotype of the cDC1 and CD8 T cell infiltrate in AM and NACM (Figure 2, Figure 3, Figure 4, Figure 5 and Figure 6). 

To evaluate the abundance of cDC1s, we stained samples for mIF analysis with antibodies that recognize the cDC1 classical markers CD11c, HLA-DR, and BDCA-3 [21,22]. We also evaluated their proximity to tumor cells by staining the HMB45 melanoma marker. Figure 2A shows the scanning of a mIF image of a whole melanoma specimen. We observed that cDC1s are distributed near the epidermis and close to the areas of tumor cells throughout the entire tissue; this can be further appreciated in a zoomed-in area (lower panel). As cDC1s were observed throughout the tissue, we selected three highly infiltrated areas, the standard of histological examination, to in-depth estimate the level of infiltration. Figure 2B shows representative H&E and mIF images of the cDC1s infiltrating stages I and III of the AM and NACM. In Figure 2C, we estimated the absolute numbers and percentages of cDC1s per 1 × 10^5^ μm^2^ finding similar abundance and proportion of these cells in both types of melanoma. We also separately compared the percentages of cDC1s in AM and NACM patients at early (I/II) and advanced (III/IV) stages and found no differences (Figure 2D). Nevertheless, when we assessed the percentage of cDC1s in early versus advanced stages, we found that early stages exhibit higher percentages of cDC1s than advanced stages (*p* = 0.0095) (Figure 2E). These data show that cDC1s are present at similar densities and distributions in AM and NACM samples. Moreover, the enrichment of cDC1s at early stages supports the idea that these cells are important for activating adaptive immune cells that control disease progression.

### 2.3. Acral Melanoma Outperforms Other Cutaneous Melanomas in the Level of Infiltration of CD8 T cells

CD8 T cells are activated by cDC1s in the tumor stroma or nearby lymph nodes. In previous work, we reported the presence of CD3 positive T cells in the immune infiltrate of AM and NACM patients [54]. Seeking to extend this observation, we compared the abundance of CD8 T cells in the stroma of AM and NACM. We first show the whole scanning of a stage III AM specimen, stained for CD8 and MART-1 (Figure 3A). MART-1 and HMB-45 are commonly used to identify melanoma cells, and we have never found differences in the staining of both markers (not shown). We observed extensive regions of CD8 T cell infiltration nearby the tumor, supporting that AM is also infiltrated by CD8 T cells that closely interact with the tumor cell. This can be better appreciated in a zoomed-in area of the tissue (the lower panel). Figure 3B shows representative images of areas of high CD8 T cell infiltration in stage I and III melanomas. Remarkably, although we did not observe significant differences in the absolute numbers of CD8 T cells per 1 × 10^5^ μm^2^ across the whole tumor specimen (Figure 3C), when we analyzed the percentage of CD8 T cells, we found a higher percentage of these cells in AM than in NACM (*p* = 0.0002). The percentage of CD8 T cells was still higher when we only considered early (I/II) (*p* = 0.04) or advanced (III/IV) (*p* = 0.001) stages (Figure 3E). Here, similar to cDC1s, we also observed a reduced infiltration of CD8 T cells in the more advanced melanoma, probably explaining tumor progression. Altogether, these data support the notion that cDC1s and CD8 T cells are also infiltrating AM, with CD8s significantly more represented in AM than NACM. Thus, these data also argue that AM bears a sufficient neoantigen load to induce anti-tumor immune responses. A substantial reduction in the level of infiltration of these immune cells marks progression to advanced tumor stages in both types of melanoma.

### 2.4. The PD-1/PD-L1 Axis Is Present in Acral Melanoma

A few studies support the idea that AM has more aggressive behavior and a poorer prognosis than NACM. Lack of neoantigen expression, and poor immune responses have been proposed as the main culprits of the AM aggressive behavior [55,56,57]. Since our data support similar levels of infiltration of two of the most central antitumor immune cells, a lack of cDC1s and CD8 T cells would not explain the AM aggressive behavior. Considering that PD-1 and PD-L1 are important inhibitory checkpoint receptors whose abundant expression explains immune cell exhaustion, tumor progression and aggressiveness, we evaluated the expression of both molecules in AM and NACM. Furthermore, these molecules are the targets of immunotherapeutic approaches that are significantly improving the survival of patients with NACM and other types of cancer [58].

Whole slide scanning of a stage IV AM and NACM allowed us to observe the expression and tissue distribution of PD-1 and PD-L1 (Figure 4A). Tumor cells are distinguished by MART-1 expression to appreciate their connection with both molecules. A zoomed-in region of the tumor illustrates that PD-1 and PD-L1 are highly expressed in both types of melanoma (right panels). Figure 4B shows representative mIF images of stage I and stage III melanoma-infiltrated areas. We used these infiltrated areas to evaluate the expression of PD-1 and PD-L1. We observed that PD-1 expression is restricted to the infiltrated areas, whereas PD-L1 is expressed both in the infiltrating and tumor cells, but it is especially highly expressed in the latter. We also found that the percentage of PD-1 and PD-L1 positive cells was similar in both types of melanoma at early (I/II) and late stages (III/IV) (Figure 4C,D). These data suggest that the tumor is controlling the immune cells-antitumor effector function. Notably, the similar expression of PD-1 and PD-L1 in AM and NACM strongly suggests that patients with AM would also benefit from anti-PD-1 or anti-PD-L1 immunotherapy, similar to the benefit exhibited by NACM.

### 2.5. Acral Melanoma CD8 T cells and cDC1s Express PD-1 and PD-L1

Since we observed that PD-1 and PD-L1 were expressed by tumor infiltrating cells, we evaluated their expression on CD8 T cells and cDC1s, particularly on advanced melanomas, to explain tumor progression. To this end, we performed multiplexed IF on five advanced AMs and five advanced NACMs. We observed cDC1s expressing PD-L1 and CD8 T cells expressing PD-1 in both types of melanoma (Figure 5A). We evaluated the percentage of cDC1s that were PD-L1 positive, finding a similar abundance of these cells in AM (28.15%) and NACM (30.74%) (Figure 5B). We also evaluated the normalized level of expression of PD-L1 by the mean fluorescence intensity (MFI), finding that cDC1s from AM exhibited two main peaks, one with low expression (MFI = 0.63) and one with high-expression (MFI = 1.63). cDC1s from NACM were more homogeneous, with a prominent intermediate peak of expression (MFI = 1.23), but also with a less abundant peak of high-expression cells (MFI = 1.83) (Figure 5D). On the contrary, the fraction of PD-1-positive CD8 T cells was significantly higher in AM (67.32%) than in NACM (56.13%) (*p* ≤ 0.05) (Figure 5C). Moreover, AM also presented a dominant peak of CD8 T cells expressing high levels of PD-1 (MFI = 1.43) (Figure 5E). We also observed that the majority of the PD-1^high^ CD8 T cells coexpressed TIM-3, supporting the hypothesis that these cells are exhausted cells (Appendix A).

We assessed the spatial relationship between PD-L1^+^ cDC1s and PD-1^+^ CD8 T cells to evaluate whether the expression of these checkpoint proteins could enable interactions between these two immune cells. Figure 5F shows examples of the mIF images used to construct an XY-positional plot representing the tissue distribution and possible connection of PD-L1^+^ cDC1s and PD-1^+^ CD8 T cells. We generated a density contour map reflecting areas with more confluence of cDC1s and CD8s. These contour maps enabled us to observe areas of close interaction between cDC1s and CD8 T cells in both types of melanoma. Although CD8 T cells are more abundant in AM, they seem to have an exhausted phenotype. Despite the high expression of the PD-1/PD-L1 axis, both AM and NACM seem to allow close interactions between cDC1s and CD8 T cells.

### 2.6. Exhausted PD-1 Positive CD8 T cells Preserve Functional Capabilities

Studies support the idea that some PD-1^+^ CD8 T cells still withhold effector functions [59,60]. Different clusters of exhausted PD-1^+^ CD8 T cells have been observed in the stroma of NACM, some with a non-functional exhausted phenotype, but others with evidence of functionality and expansion capacity. The latter population has been shown to further expand upon treatment with anti-PD-1 antibodies [61,62,63,64]. We then evaluated whether CD8 T lymphocytes expressing PD-1 still retained functional characteristics assessing the expression of IFN-γ and KI-67. Figure 6A shows representative H&E and mIF images of the five AM and NACM advanced melanomas. We observed PD-1^+^ CD8 T cells expressing both functional molecules in both types of melanoma. We selected the PD-1^+^ CD8 T cells and evaluated the proportions of cells expressing these functional markers. Interestingly, we observed a predominance of KI-67 positive cells in AM compared with NACM (49.81% and 36.59%, respectively; *p* < 0.05). Of note, we observed an IFN-γ expressing subset of PD-1^+^ CD8 T cells in both melanomas, with NACM (40.31%) showing a non-significant predominance of this population (AM = 35.91%) (Figure 6B). In general, we observed a similar abundance of functionally-exhausted CD8 T cells in both types of melanoma (Figure 6C). The MFI of KI-67 expression was also similar (Figure 6D), but IFN-γ was clearly diminished in PD-1^+^ CD8 T cells of AM (Figure 6E). Therefore, advanced AM exhibits a lower proportion of IFN-γ-expressing PD-1^+^ CD8 T cells, and reduced levels of IFN-γ expression compared with advanced NACM. Importantly, we observed a small subset of PD-1^+^ CD8 T cells with high expression of IFN-γ on AM (MFI = 1.43). These data suggest that both types of melanoma withhold functional PD-1^+^ CD8^+^ T cells at advanced stages, although in AM these cells are preferentially in a state of proliferation and producing low amounts of IFN-γ. These cells may still be active and could be rescued by anti-PD-1 immunotherapy.

### 2.7. Flow Cytometry Analysis Validates the Presence of Functionally Exhausted CD8 T Cells in Acral Melanoma 

To validate the mIF and machine learning workflow results, we conducted a multiparametric flow cytometry analysis. We obtained the immune cells from a fresh metastatic sample of AM and a fresh skin biopsy from a healthy donor and evaluated the expression of memory (CD45RO), exhaustion (PD-1), activation (CD69), and function (IFN-γ & KI-67) markers on CD8 T cells. We confirmed the existence of CD8 T cells co-expressing these molecules. We also observed that the healthy skin control was highly infiltrated by PD-1^+^ CD8 memory T cells (58.1% of CD8 T cells in the AM sample vs. 65.6% in the control skin) (Figure 7A). We were surprised by this observation, as it may indicate that PD-1 expression is not only associated with exhaustion but also with the normal surveillance function of the immune cells in peripheral tissues, such as the skin. An extended phenotype of these cells showed high expression of two other exhaustion markers, CD39 and TIM-3 (Appendix A) [63,65,66,67]. To assess whether the PD-1^+^ CD8 memory T cells from the healthy skin and the AM were functionally equivalent, we performed a non-supervised t-distributed stochastic neighborhood embedding (t-SNE) clustering. We observed non-overlapping clusters between CD8 T cells infiltrating AM and control skin (Figure 7B), supporting differences in the CD8 T cells derived from each tissue. PD-1^+^ CD8 memory T cells derived from the AM appeared more functional, expressing considerably more IFN-γ and KI-67, while those derived from the healthy skin were CD45RO^high^, and both expressed roughly similar levels of CD69 and PD-1 (Figure 7C). The t-SNE unbiased analysis revealed the presence of eight clusters (C1 to C8), with AM T cells appearing more homogeneous since only two of the clusters were derived from this tissue (C6 and C8) (Figure 7D,E). The melanoma clusters were mainly distinguished by the level of expression of IFN-γ (C8), and KI-67, CD69, and PD-1 (C6) (Figure 7D,F). Summarizing the data, we showed evidence of a robust infiltration of relevant antitumor immune cells in AM. Both immune and tumor cells expressed exhaustion markers, but despite this phenotype, PD-1+ CD8 T cells showed evidence of an ongoing effector function demonstrated by the expression of IFN-γ and KI-67. 

## 3. Discussion

Melanoma is an important health problem worldwide. In Mexico, melanoma has grown by 500% in incidence and by 115% in mortality in recent years [2,68]. The GCO estimates that melanoma incidence and mortality will continue to grow throughout Latin America, highlighting the importance of studying this disease in the Mexican and Hispanic populations [54]. With this scenario, we conducted this study with patients from a national healthcare institute in Mexico (IMSS) that cares for about half of its inhabitants [69]. We observed a high prevalence of AM that accounted for 73.9% of all patients in the cohort. This result is consistent with AM being the most prevalent melanoma in Hispanics. We even observed a higher proportion of this type of melanoma in our cohort than the ones reported in Hispanics from USA studies (37.94%) [10]. Furthermore, the proportion of AM in our cohort was even higher than others reported for the country (44% for Mexico’s National Institute of Cancer (INCAN)) [2], perhaps due to the overrepresentation of the non-Caucasian population that is cared for by IMSS. Indeed, a previous report from the private ABC hospital only found 17% of AM [70]. We also noticed that AM, probably due to its occult location, is more likely to be diagnosed at advanced stages than NACM from the same cohort. 

NACM has shown the greatest rate of single nucleotide variations (SNV), arguably making it the most immunogenic type of cancer [13,16,71]. On the contrary, AM exhibits a reduced rate of SNVs due to its origin in areas of the skin that are usually shielded from exposure to UV light. This has led to the hypothesis that AM is poorly immunogenic since SNV mutations are needed to generate the neoantigen load that triggers antitumor responses. Furthermore, AM lacks proper infiltration of antitumor immune cells [53,72,73]. Due to the sparse research on AM, these preconceived ideas have not been substantially challenged. We have recently compared the mutational load of AM and NACM, confirming its reduced level of SNVs [54]. However, we also noticed that AM carries an SNV load comparable to other neoplasms known to be immunogenic and responsive to immunotherapeutic regimes. 

In this study, we demonstrated in situ, on a proteomic level, that AM is infiltrated by cDC1s and CD8 T lymphocytes at similar and even higher levels than NACM. These data argue that AM is also an immunogenic neoplasm. On the one hand, these results suggest that the AM stroma is enriched with damage-associated molecular pattern (DAMPs) molecules that induce a robust infiltration of dendritic cells, particularly of cDC1s. These cells are recognized as the more competent cells to activate CD8 T cells because of their capacity to cross-present tumor-derived antigens on MHC class I molecules [74,75]. On the other hand, there is probably a significant expression of tumor-derived antigens that can be recognized by cytotoxic CD8 T cells. Indeed, throughout all the mIF images acquired, it was clear that both cell types are present near areas enriched with tumor cells. Some studies support the idea that AM is inherently more immune-silent than NACM. For instance, Li Jiang and colleagues observed by single-cell RNA-sequencing that AM was less infiltrated than NAMC, and the infiltrate presented a more exhausted phenotype [52]. Our data differ from those presented in this study. Although the origin of this discrepancy between studies is unclear, it could be explained by the different transcriptomic and proteomic approaches used.

We also observed that the density of cDC1s and CD8 T cells decreases at advanced melanoma stages, suggesting that these cells are critical to controlling tumor progression and that they are interacting with each other to eliminate tumor cells efficiently. The reduction in antitumoral immune cells at advanced melanoma stages could be explained by a sustained immuno-edition process carried out by cancer cells as a mechanism to escape the immune response, but also by other mechanisms of immunosuppression [76,77,78]. Today, one of the most studied mechanisms of immunosuppression is the expression of exhaustion markers, such as the checkpoint molecules PD-1/PD-L1 [79,80,81]. We found that PD-L1 and PD-1 are abundantly expressed in AM. Both the tumor cells and the immune infiltrate express the former, while the latter better marks immune cells. Most of the novel therapies in use or in clinical trials are inhibitors of this axis [45,58,82,83,84,85,86,87,88]. Antibodies directed against these molecules are the first line of treatment for patients with advanced unresectable melanoma in high-income countries where NACM is highly prevalent [89,90].

The success of anti-PD-1 therapy has been associated with the specific expression of its target PD-1 on exhausted CD8 T cells, leading to the belief that these cells are prone to recover their effector function upon immunotherapy. This has been confirmed by an anti-PD-1-induced recovery of expression of effector molecules, increased proliferation, and expansion of the T cells’ lifespan [43,61]. Our data support the finding that AM and NACM have a similar antitumor response in terms of cDC1 and CD8 T cell abundance, and both types of melanomas exhibit similar expression levels of PD-1 and PD-L1. This is important since these data argue that PD-1 blockade would also benefit AM patients. 

Recent studies support the notion that PD-1 expression is not exclusive to exhausted cells, since PD-1 expression has been observed in newly activated functional T lymphocytes [59,60]. This has led to the proposal that there are different subpopulations of PD-1^+^ CD8 T cells, some of which have progenitor, effector, and proliferative capacities, while others are non-functional and exhibit evidence of mitochondrial dysfunction and apoptotic programs in play [30,32,38].These T cell subpopulations may be distinguishable by their PD-1 expression levels. Thus, PD-1^high^ CD8 T lymphocytes are probably more enriched with less functionally exhausted cells, while PD-1^low/mid^ CD8 T cells are enriched with cells that still maintain active effector functions and are the target for anti-PD-1 immunotherapy rescue. Indeed, we observed by mIF that the PD-1^high^ CD8 T cells are also co-expressing TIM-3 (Appendix A), and by flow cytometry, the PD-1^+^ CD8 T cell clusters found in melanoma also co-express TIM-3 and CD39 (Appendix A). In this regard, in situ functional evaluation of PD-1^+^ CD8 T cells demonstrated the presence of functional effector (IFN-γ^+^) cells and of expanding (KI-67^+^) cells. We observed subtle differences between AM and NACM, for instance, there are more PD-1^+^ CD8 T cells expressing KI-67 in AM, while in NACM, PD-1^+^ CD8 T cells expressed higher levels of IFN-γ than in AM. Importantly, we also observed a small subset of PD-1^+^ CD8 T cells in AM that expressed IFN-γ at similar levels to the NACM PD-1^+^ CD8 T cells (see the population with a MFI = 1.43 in Figure 6E). We consider that this population could be targeted by immunotherapy, promoting its further expansion and effector function. 

As a proof of concept, we mapped areas of proximity between PD-1^+^ CD8 T cells and PD-L1^+^ cDC1s in the stroma of one of the AM and NACM samples. In both types of melanoma, areas of colocalization of both populations were found, arguing that both cells are interacting. At this time, we cannot discern whether these interactions are immunosuppressive or activating. In future studies, we should analyze a larger number of samples to obtain more robust estimations, and we should also include an extended panel of activating and exhaustion markers to better discriminate the interactions between *bona fide* effector cells and non-functional exhausted cells. We also used flow cytometry and unsupervised clustering with t-SNE maps to confirm the presence of PD-1^+^ CD8 T cells that co-express IFN-γ and KI-67. This analysis allowed us to observe a cluster (C6 of Figure 6) of AM PD-1^+^ CD8 T cells that expressed IFN-γ and KI-67 but also the memory marker CD45RO and high levels of CD69, an early activation and tissue residency marker. We also found a less abundant cluster (C8) showing high expression of IFN-γ and CD45RO but low to no expression of PD-1, CD69, and KI-67, probably representing effector memory T cells. These clusters were not present in healthy skin.

Taken together, these data show that AM is highly represented in a cohort of patients from a native Hispanic country. Importantly, AM is highly infiltrated by antitumor immune cells such as cDC1 and CD8 T lymphocytes. Although both cell types express the PD-1 and PD-L1 markers of exhaustion, we provide evidence that PD-1^+^ CD8 T cells from AM withhold functional capabilities, as they express IFN-γ and KI-67. AM tends to be diagnosed later than NACM, and advanced melanoma stages are characterized by a decreased infiltration of cDC1s and CD8 T cells. For the reason that we observe the presence of IFN-γ^+^ and KI-67^+^ PD-1^+^ CD8 T cells in stage IV AM, our study argues that these cells could be rescued and their effector functions recovered by the immunotherapeutic regimes commonly used to treat advanced NACM. However, a deeper phenotypic characterization of these populations is needed to fill the knowledge gap that prevails in AM. In Mexico and other low- or medium-income countries, chemotherapy and surgical resection remain the most widely used therapies to treat advanced melanoma, despite their poor clinical benefit [3]. Overall, our data argue that the aggressiveness observed in AM is mostly a result of late diagnosis and limited use of therapeutic options. Intensive research in the AM immune stroma is needed to continue deciphering its ability to activate protective immune responses and to provide better therapeutic options to patients.

## 4. Materials and Methods

### 4.1. Human Melanoma and Control Samples

Thirty-eight paraffin-embedded tissues from resected melanoma and the corresponding H&E staining were obtained from the Department of Pathology of the Oncology Hospital CMN-SXXI IMSS. The Mexican Institute of Social Security (IMSS) is the prime public health provider in Mexico. IMSS cares for Mexican workers and their families employed by non-state companies, mainly the low- and middle-income population, ultimately serving close to half of the Mexican population [69]. Health care at the Institute is divided into three levels: primary, secondary, and tertiary care. Tertiary care treats highly complex diseases requiring specialized equipment, facilities, and treatments, such as cancer. The IMSS Oncology Hospital is one of the most important hospitals that provide attention to cancer patients from Mexico City and nearby states from the central region of the country.

Based on the location of the primary lesions, patients were classified as AM (n = 30) and NACM (n = 8). Other clinical parameters were obtained from the medical history files of the patients. As NACM samples were underrepresented in the IMSS cohort, we selected five NACM patients from other hospitals. Three were obtained from the Cancer Center of Hospital Centro Médico ABC, a private hospital in Mexico City, and two were obtained from UMAE General Manuel Avila Camacho (also IMSS). See Appendix A for a summary of the patient’s demographic and clinical data. The protocol was approved by the IMSS and ABC Institutional Scientific and Ethics Board of Reviews, with protocol numbers R-2019-785-05 and ABC-21-39, respectively.

Fresh melanoma tissue was obtained from a resection surgery at the Department of Pathology of the Hospital de Oncología CMN-SXXI IMSS. Tissue was received in saline solution and then processed for dermal cell isolation. Control skin biopsies (without disease) were obtained from skin samples during planned gastrointestinal surgeries performed at Hospital de Especialidades CMN-SXXI IMSS. Control subjects did not present any malignant, autoimmune, or inflammatory processes. The collection of fresh samples was conducted under the approval of the IMSS Institutional Scientific and Ethics Board (protocol number PI-2019-7363), and patients and donors signed an informed consent agreeing to participate in the study.

### 4.2. Isolation of Dermal Cells from Skin Biopsies

Biopsies from melanoma patients and control subjects were placed in supplemented RPMI 1640 (Biowest, Nuaillé, France) and supplemented with fetal bovine serum (FBS) 10%, HEPES (Biowest, Nuaillé, France), penicillin/streptomycin (Biowest. Nuaillé, France)) non-essential amino acids, glutamine (Gibco, Bleiswijk, Netherlands) ciprofloxacin (Laboratorios Senosianin, Celaya, Mexico) and β-mercaptoethanol (Sigma Aldrich, St. Louis, MO, USA) and dispase II (Grade II protease (Roche, Penzberg, Germany)) overnight at 4°C. Furthermore, the epidermis was separated from the dermis. The dermal segments were cultured in supplemented RPMI 1640 (37°C and 5% CO_2_) to allow the migration of the cells to the medium due to a nutrient concentration gradient. Cells were harvested after seven days of culture and cryopreserved to perform further analysis.

### 4.3. Whole Tissue Scanning

To obtain images of the complete melanoma specimen in H&E or IF, we used an Aperio CS2 (Leica Biosystems, Wetzlar, Germany) with a 20x objective for the former and an epifluorescence slide scanner Aperio FL (Leica Biosystems) with a 20x objective for the latter. Image export of regions of interest was performed with eSlide Scan Scope v12.3.3 (Leica Biosystems). The images were analyzed using FIJI (ImageJ NIH, Bathesda, Maryland, AR, USA) software v2.3.0/1.53t. 

### 4.4. Tissue Embedding and Immunofluorescence Staining

Resected tissue was placed in tissue capsules and preserved in formalin for 12–24 h. Furthermore, the tissue was dehydrated with an Ethanol/xylene solvent train (H2O, Et50%, Et80%, Et100%, Et50%/Xylene50%, Xylene). To preserve the tissue, resection products were infiltrated with paraffin for 1–2 days, and finally the tissue was embedded into blocks. 

Tissue sections of 3 μm were cut with a microtome and mounted on glass slides (Superfrost Plus Green, EMS, Hatfield, PA, USA). Slides were placed for 45 min in an oven (70°C) to remove excess paraffin. Tissues were rehydrated with a Xylol/Ethanol train of solvents. Antigen retrieval was performed using citrate buffer pH 6.0 (sodium citrate 10 μM) at 90°C for 20 min. Furthermore, samples were permeabilized with a solution of 10 mg/mL bovine serum albumin, 5% horse serum, 0.02% sodium azide, and 0.3% Triton (Sigma-Aldrich St. Louis, MO, USA) for 2h. Following permeabilization, samples were incubated overnight with the corresponding primary antibodies: anti-CD8, anti-MART, anti-CD11c, anti-HLA-DR, anti-BDCA3, and anti-KI-67. Primary antibodies were revealed with secondary conjugated antibodies: anti-rabbit Alexa Fluor 488 (711-547-003, Jackson Immunoresearch, West Grove, PA, USA), anti-rat Alexa Fluor 594 (712-585-153, Jackson Immunoresearch), anti-mouse Alexa Fluor 647 (715-605-151, Jackson Immunoresearch). When specified, dye-conjugated antibodies anti-CD8, anti-PD1, anti PD-L1, and anti-IFN-γ were incubated overnight. Appendix A includes information on the antibodies used in IF assays. After that, nuclei were counterstained with Hoechst (Invitrogen Waltham, MA, USA) for 10 min. Sections were mounted with 10% glycerol in PBS. Images were acquired after this step of staining and used for analysis. To evaluate more than four markers, mIF was performed according to a protocol previously reported (see Section 4.6). To perform the second round of staining, coverslips were removed by soaking the slides in 1X PBS. Samples were acquired as described below. Using the H&E scans, three highly infiltrated areas close to the tumors were chosen to assess the specified phenotype of the cells.

### 4.5. Confocal Microscopy

Micrographs were obtained on a Nikon Ti Eclipse inverted confocal microscope (Nikon Corporation, Minato, Tokyo, Japan) using NIS Elements v.4.50. Imaging was performed using a 20x (dry, NA 0.75) objective lens. Additional magnification (3.4x) was attained through Nyquist’s sampling during image acquisition. Three areas of high immune infiltrate from each group of patients and controls were taken to quantify the density of CD8 T cells and cDC1s. When multiplexed immunofluorescence was performed, the same areas were evaluated for each cycle in order to determine the colocalization of labels. Automated image overlapping and further analysis were carried out using FIJI ImageJ Software (ImageJ NIH, Bathesda, Maryland, AR, USA) [91].

### 4.6. Immunofluorescence Image Analysis

The IF analyses were performed using our machine learning method described in [50]. This method involves automatic nuclei segmentation using our model trained in Convolutional Neuronal Network Cellpose. The model subsequently measures the expression of different markers to define the phenotype with Annotater in ImageJ. The results obtained were analyzed using Python scripts. The percentage of positive cells was calculated by dividing the total number of cells positive for a given phenotype by the total number of cells in the field multiplied by 100. The density was measured by dividing the number of cells of interest by the total area of the field. The results shown in the graphs represent the median value of three analyzed areas per sample or patient. Although fibrotic areas and blood vessels can exhibit high autofluorescence, these areas were not included in the experimental assessments and comparisons between AM and NACM, for which only marks with nuclei were analyzed. In the case of the XY coordinate maps and density contour maps, the information obtained with our machine learning image analysis method was plotted using the CytoMap software V.1.4.20 [92]. 

### 4.7. Flow Cytometry Staining

Cryopreserved cells were obtained at step 4. Two were thawed and then washed twice using PBS buffer and blocked with FACS (Fluorescent Activated Cell Sorting) buffer. Cell surface staining was then performed by adding the following antibodies: anti-CD45-PE/Cy7, anti-CD45RO-PE/Dazzle94, anti-CD8-APC/Cy7, anti-CD4-BV650, anti-CD69-BV750, anti-PD-1-PE, anti-TIM3-APC, anti-CD39-BV711 (Biolegend, San Diego, CA, USA, Clone: A1), anti-TIM3-APC (ThermoFisher, Waltham, MA, USA, Clone: F38-2E2). Live/Dead Fixable Violet staining was included. Cell surface-stained cells were washed twice with PBS buffer, and intracellular staining was performed using the True-Nuclear Transcription Factor buffer set (Biolegend) according to the manufacturer’s instructions. Intracellular staining included: anti-IFN-γ-BV510, and anti-KI-67-BV605. Appendix A includes information on the antibodies used in flow cytometry assays. Cells were acquired in a BD Influx cytometer; the stopping gate was set at 10,000 CD45RO^+^ events; 100,000 and 245,000 total events were acquired in skin control and TILs, respectively. Files were analyzed using Flowjo v10.8 software (Becton Dickinson, Franklin Lakes, NJ, USA).

Unsupervised clustering was C. Briefly: lymphocytic, singlets, live, CD45RO, and CD8 positive events were down-sampled and exported as Flow Cytometry Standard (FCS) files. CD8 FCS files were concatenated (18699 events), and t-SNE clusterization was performed with IFN-γ, PD-1, KI-67, and CD69. We used 1000 iterations, a perplexity of 30, a learning rate of 7000, the exact KNN (K-Nearest Neighbors) algorithm, and the Barnes-Hut gradient algorithm [93] to generate an unsupervised clustering map with the Phenograph plugin using the same compensated parameters [94]. Heat maps and representative clusters were obtained with the Cluster Explorer plugin.

### 4.8. Statistical Analysis

All data are presented as the mean ± SD (standard deviation). We evaluated the normal (Gaussian) distribution of all the data with the D’Agostino test. For experiments with three groups, ANOVA with post hoc multiparametric comparison was chosen to compare the different groups (Figure 2 and Figure 3). For the case of two groups of independent quantitative variables, we applied the unpaired Student’s *t*-test (Figure 4, Figure 5 and Figure 6). All statistical analyses were performed using Prisma software (GraphPad 8). Statistical significance was defined as * *p* < 0.05, ** *p* < 0.01, *** *p* < 0.001 and **** *p* < 0.0001.

## Figures and Tables

**Figure 1 ijms-24-04786-f001:**
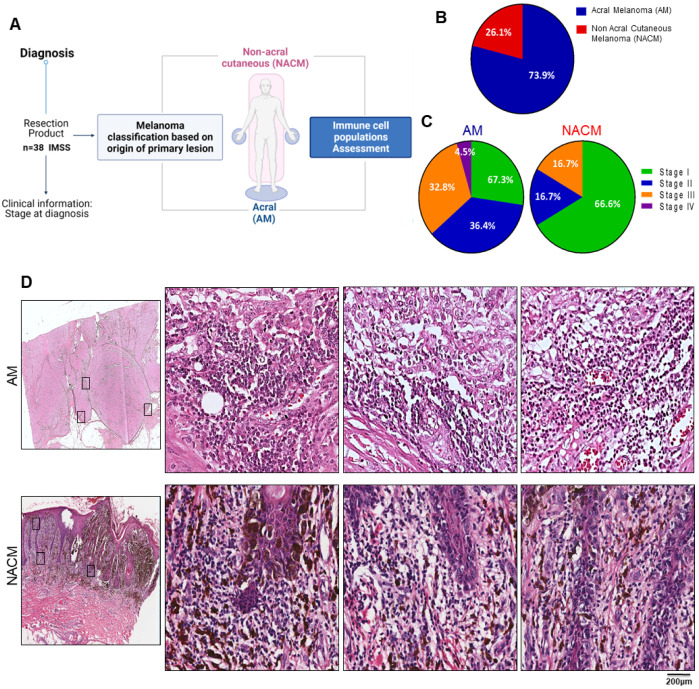
Characteristics of the original IMSS study cohort. (**A**) Schematic representation of the workflow of the melanoma samples. Tissue resection specimens and clinical information were obtained at diagnosis from The Mexican Institute of Social Security (IMSS). The study impartially covered archival samples derived from patients admitted during the years 2017–2018. Classification of the type of melanoma was based on the anatomical origin of the primary lesion. Skin lesions from nail beds, palms, and soles were classified as acral melanomas (AM), while lesions originating from other skin locations were non-acral cutaneous melanomas (NACM). (**B**,**C**) Pie charts showing the proportion of patients with AM and NACM (n = 38) found in the original IMSS cohort (**B**), and (**C**) the proportion of each melanoma classified according to the staging system. (**D**) Representative images of hematoxylin and eosin (H&E) staining of AM (top) and NACM (bottom) showing the whole resection slide (left panel), and three digital zooms (hollow squares) of areas that are highly infiltrated (right panels).

**Figure 2 ijms-24-04786-f002:**
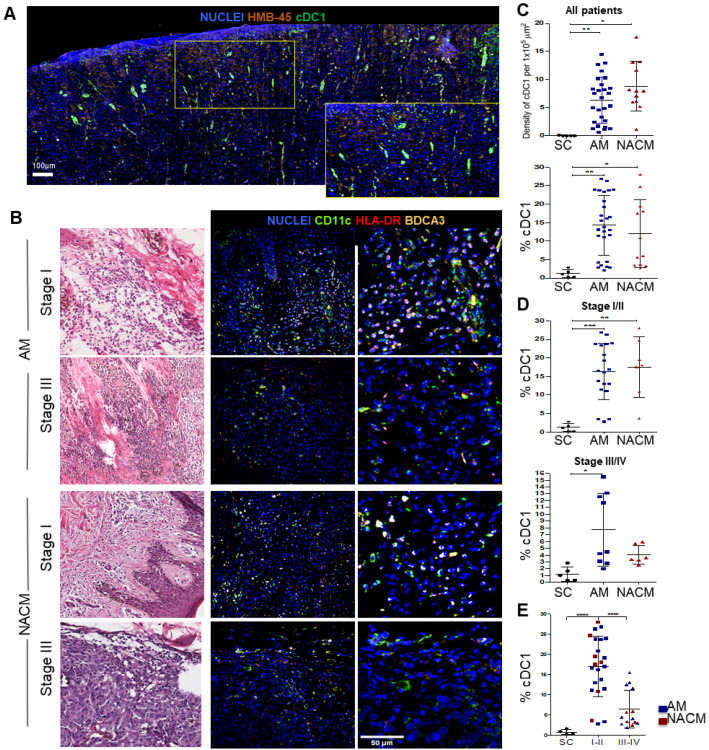
Assessment of the melanoma infiltrating conventional type 1 dendritic cells (cDC1). (**A**) Whole slide scanning of a representative advanced stage III acral melanoma (AM) showing the global distribution of CD11c^+^ HLA-DR^+^ BDCA3^+^ cDC1s (green) near the HMB-45^+^ tumor cells (orange). Digital zoom is also presented for a better appreciation (yellow panel). (**B**) Representative hematoxylin & eosin (left column) and immunofluorescence (IF) (middle and right columns) images of stage I and III AM and non-acral cutaneous melanoma (NACM). The right column represents a digital zoom of a highly infiltrated area. IF images were stained with CD11c (green), HLA-DR (red), BDCA3 (yellow) and Nuclei (Blue). (**C**,**D**) Plots with the following comparisons between AM and NACM, (**C**) the absolute numbers of cDC1s per 1 × 10^5^ μm^2^ (top) and percentages of cDC1s (bottom); (**D**) early stages I and II (top) and advanced stages III and IV (bottom). (**E**) Plot comparing the percentage of cDC1s between early and advanced stages. Percentages of cDC1s were estimated among all nucleated cells in the highly infiltrated areas selected. Appendix A shows the autofluorescence control for this experiment. For this analysis, we used 38 samples from the IMSS cohort plus five advanced NACM samples from other hospitals: AM (blue, n = 30) and NACM (red, n = 13). Healthy skin control (SC) samples were obtained from surgical resections unrelated to cancer (black; n = 8). Statistical analysis: One Way ANOVA’s test with Tukey’s post hoc test for multiple comparisons. * *p* < 0.05, ** *p* < 0.01, *** *p* < 0.001, and **** *p* < 0.0001.

**Figure 3 ijms-24-04786-f003:**
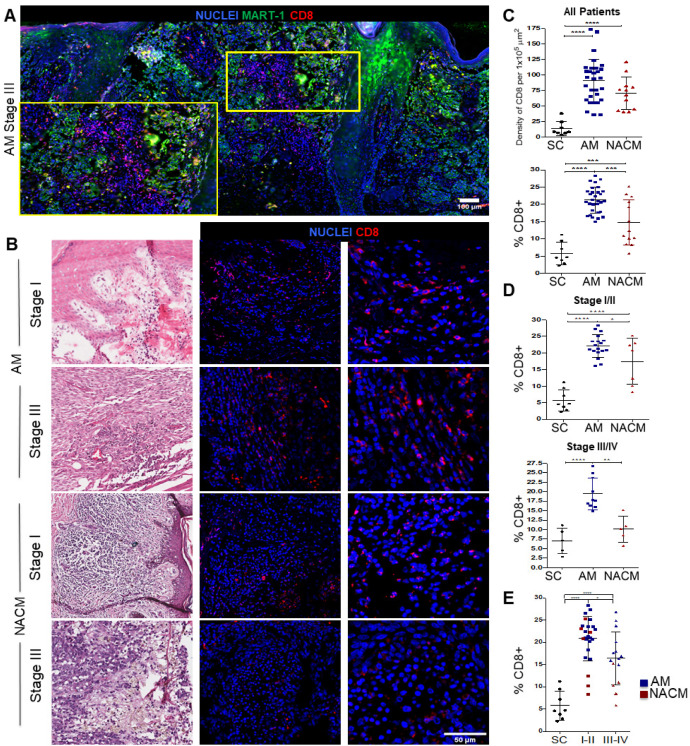
Assessment of the melanoma infiltrating CD8 T cells. (**A**) Whole slide scanning of a representative advanced stage III acral melanoma (AM) showing the global distribution of CD8 T cells (red) near the MART1+ tumor cells (green). Digital zoom is also presented for a better appreciation (yellow panel). (**B**) Representative hematoxylin & eosin (left column) and immunofluorescence (IF) (middle and right columns) images of stage I and III AM and non-acral cutaneous melanoma (NACM). The third column represents a digital zoom of a highly infiltrated area. Samples were stained for CD8 (red) and nuclei (blue). (**C**,**D**) Plots with the following comparisons between AM and NACM, (**C**) the absolute numbers of CD8 T cells per 1 × 10^5^ μm^2^ (top) and percentages (bottom); (**D**) early stages I and II (top) and advanced stages III and IV (bottom). (**E**) Plot comparing the percentage of CD8 T cells between early and advanced stages. The percentage of CD8 T cells was estimated among all nucleated cells in the highly infiltrated areas selected. Appendix A shows the autofluorescence control for this experiment. For this analysis, we used 38 samples from the original IMSS cohort plus five advanced stage NACM samples obtained from other hospitals: AM (blue, n = 30) and NACM (red, n = 13). Healthy skin control (SC) samples were obtained from surgical resections unrelated to cancer (black; n = 8). Statistical analysis: One-Way ANOVA’s test with Tukey’s post hoc test for multiple comparisons. * *p* < 0.05, ** *p* < 0.01 *** *p* < 0.001, and **** *p* < 0.0001.

**Figure 4 ijms-24-04786-f004:**
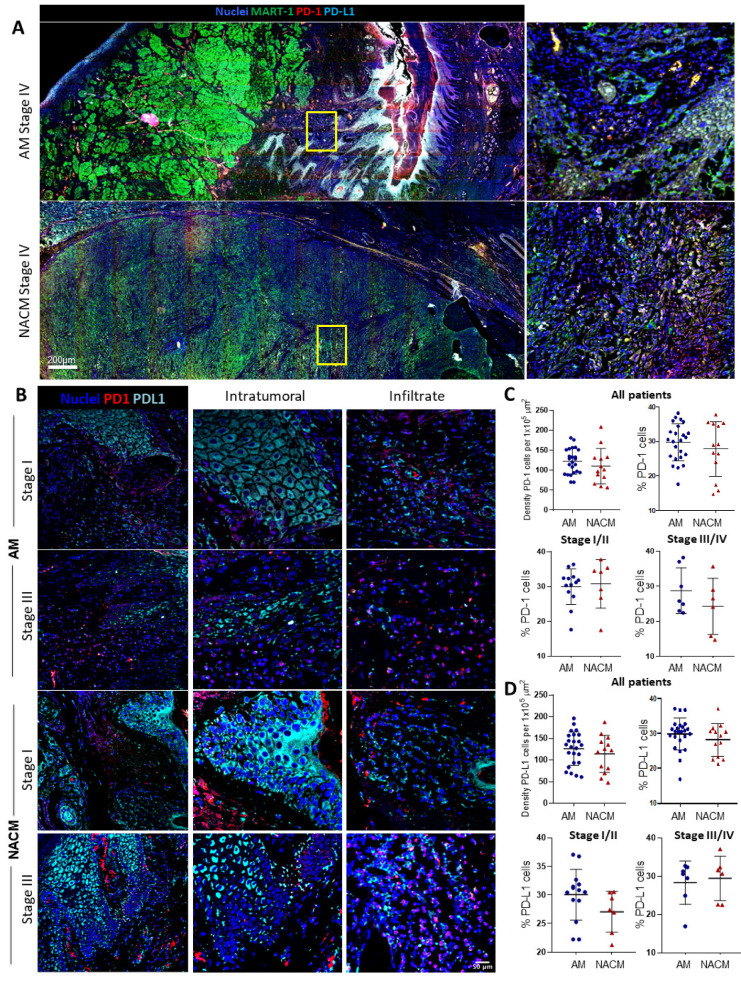
The PD-1/PD-L1 axis is active in acral melanoma samples. (**A**) Whole slide scanning of a representative immunofluorescence (IF) image of advanced stage IV AM (top) and NACM (bottom) showing the global distribution of PD-1 (red) and PD-L1 (cyan). IF images also show tumor cells marked with MART-1 (green) and nuclei (blue) and digital zooms (right panel, yellow squares). (**B**) Representative IF images of Stage I and III AM and NACM. Staining: PD-1 (red), PD-L1 (cyan) and nuclei (blue). Digital zooms show an intratumoral area (middle) and a highly infiltrated area (right). (**C**,**D**) Plots with the following comparisons between AM and NACM, (**C**) the absolute numbers of PD-1^+^ cells per 1 × 10^5^ µm^2^ (left) and percentages (right) in all patients, percentages of PD-1^+^ cells in early stages I and II (bottom left) and advanced stages III and IV (bottom right). (**D**) The absolute numbers of PD-L1^+^ cells per 1 × 10^5^ µm^2^ (left) and percentages (right) in all patients, percentages of PD-L1^+^ cells in early stages I and II (bottom left), and advanced stages III and IV (bottom right). Percentages of PD-1^+^ and PD-L1^+^ cells were estimated among all nucleated cells present in the highly infiltrated areas selected. Appendix A shows the autofluorescence control for this experiment. For this analysis, we used 38 samples from the original IMSS cohort plus five additional advanced NACM samples derived from other hospitals: AM (blue, n = 30) and NACM (red, n = 13). Statistical analysis: Unpaired Student’s *t*-test.

**Figure 5 ijms-24-04786-f005:**
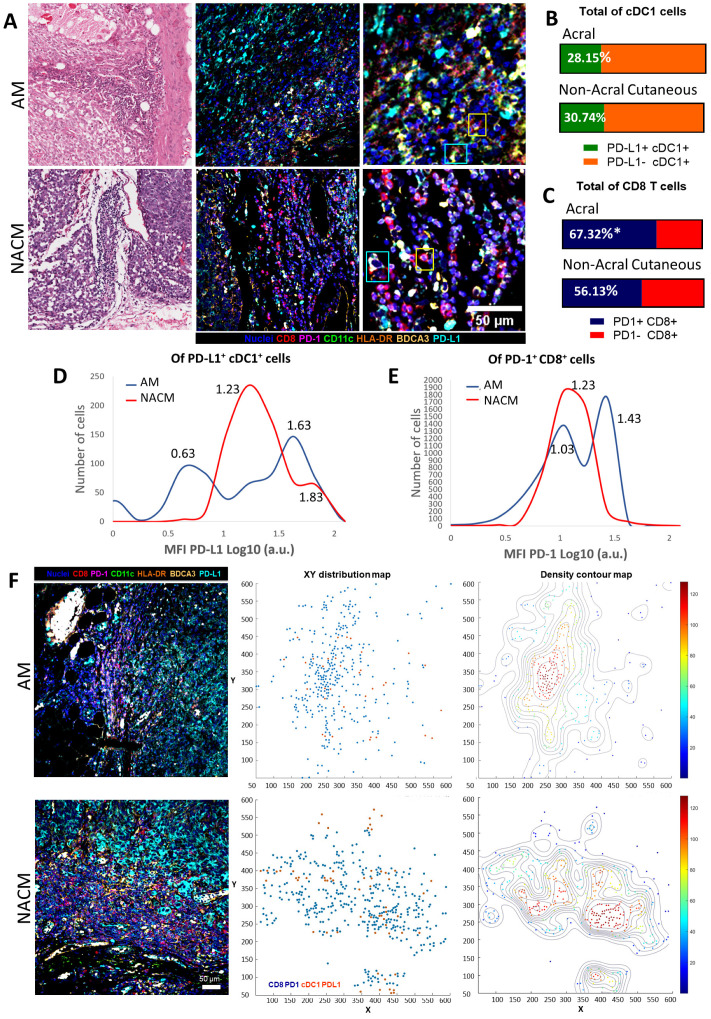
Acral melanoma is infiltrated by PD-1^+^ CD8 T cells and PD-L1^+^ cDC1s. (**A**) Representative hematoxylin & eosin (left) and multiparametric immunofluorescence (mIF) images (middle and right (zoomed of a highly infiltrated area)) of advanced acral melanoma (AM) and advanced non-acral cutaneous melanoma (NACM). IF staining: CD8 (red), CD11c (green), HLA-DR (orange), BDCA3 (yellow), PD-L1 (cyan), PD-1 (magenta) and nuclei (blue) (middle panels). Yellow and cyan squares point out to PD-1^+^ CD8 T cells and PD-L1^+^ cDC1 cells, respectively. The proportion of PD-L1^+^ and PD-L1^-^ cDC1s (**B**), and of PD-1^+^ and PD-1^-^ CD8 T cells (**C**) in AM and NACM. (**D**,**E**) Histograms showing the mean fluorescence intensity (MFI) of PD-L1 expression on cDC1s (**D**) and of PD-1 on CD8 T cells (**E**) in AM (blue histogram) and NACM (red), arbitrary fluorescence units Log10 scale. (**F**) Analysis of the spatial relationship between PD-1^+^ CD8 T cells and PD-L1^+^ cDC1s. (**F**) Representative IF images (left) were used to construct XY-Cartesian coordinate (middle) and density contour (right) maps for AM and NACM to unveil regions in the stroma of melanoma in which both types of cells were in proximity. In (**A**–**E**) comparisons between AM and NACM were made using five samples of advanced stage III and IV for each type of melanoma. We show in Appendix A a representative staining of every channel to provide a better visualization of each mark individually. In addition, we show in Appendix A the acquired images of the bleached tissue to illustrate that the first round of fluorescence staining was excluded from the analysis of the second round of staining. In (**F**) one advanced stage IV sample was used for each type of melanoma. In the XY distribution map PD-1^+^ CD8 T cells are shown in blue and PD-L1^+^ cDC1 cells in red; in the density contour plot, red dots point to areas in which both types of cells are in close interaction. Statistical analysis: Student’s *t*-test, * *p* < 0.05.

**Figure 6 ijms-24-04786-f006:**
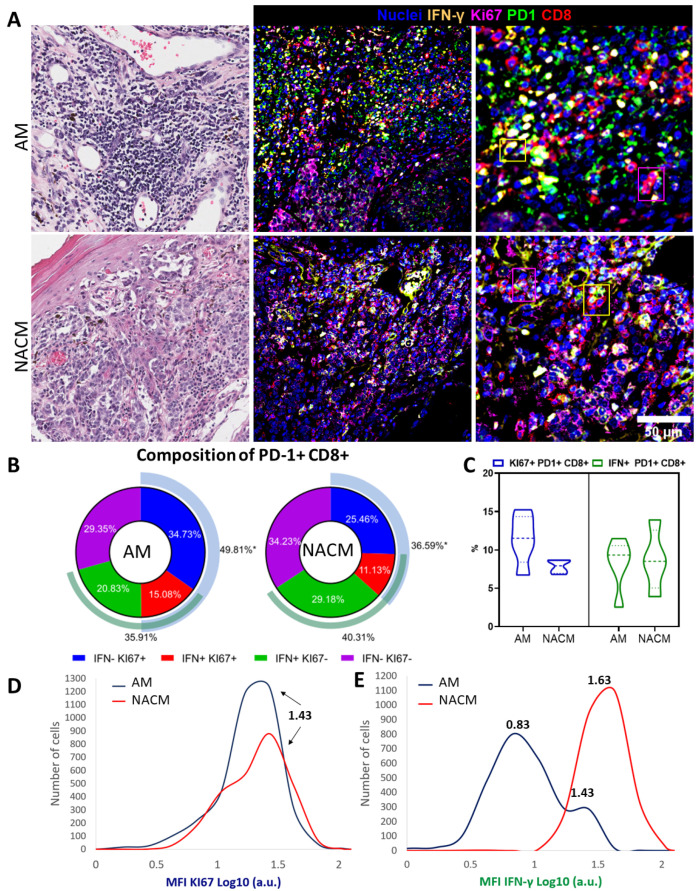
Melanoma infiltrating PD-1^+^ CD8 T cells express functional markers. (**A**) Representative hematoxylin & eosin (left) and immunofluorescence (IF) images (middle and right (zoomed of a highly infiltrated area) of advanced acral melanoma (AM) and advanced non-acral cutaneous melanoma (NACM). IF staining: CD8 (red), PD-1 (green), IFN-γ (yellow), KI-67 (magenta) and nuclei (blue). Yellow and magenta squares point out to IFN-γ^+^ PD-1^+^ CD8^+^ T cell and KI-67^+^ PD-1^+^ CD8^+^ T cell, respectively. (**B**) Pie charts showing the proportion of PD-1^+^ CD8^+^ T cells expressing IFN-γ and/or KI-67. External elements represent the total IFN-γ fraction (green), and total KI-67 fraction (blue). (**C**) Violin plots showing comparisons of the percentages of KI-67^+^ PD-1^+^ CD8 T cells (blue) and of IFN-γ^+^ PD-1^+^ CD8 T cells (green) in AM and NACM. (**D**,**E**) Normalized mean fluorescence intensity (MFI) of KI-67 (**D**) and IFN-γ (**E**) expression in KI-67^+^ PD-1^+^ CD8 T cells or IFN-γ^+^ PD-1^+^ CD8 T cells, respectively. Arbitrary fluorescence units Log10 Scale. We show in Appendix A a representative staining of every channel to better visualize each mark individually. In addition, we show in Appendix A the acquired images of the bleached tissue to illustrate that the first round of fluorescence staining was excluded from the analysis of the second round of staining. Statistical analysis: Student’s *t*-test, * *p* < 0.05.

**Figure 7 ijms-24-04786-f007:**
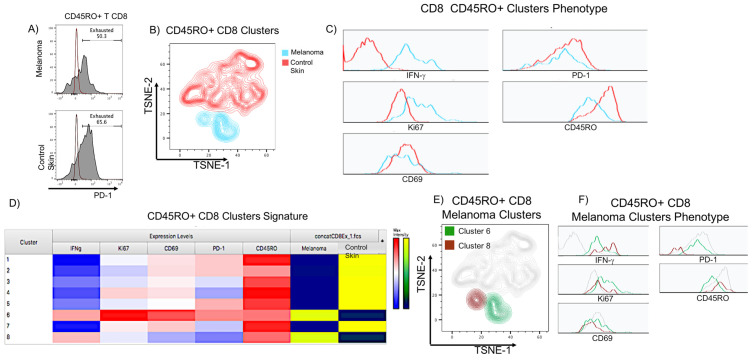
Flow cytometry analysis validates the presence of functional PD-1+ CD8 T cells in the stroma of acral melanoma. Stromal cells were obtained from one acral melanoma (AM) and one healthy skin control (SC) derived from a surgical resection unrelated to cancer. Memory CD45RO+ CD8 T cells were clustered together for further analysis. (**A**) Histogram of PD-1 expression on CD45RO+ CD8 T cells (gray histograms), the autofluorescence control is shown in maroon histograms. (**B**) CD45RO+ CD8 T cells non-supervised t-distributed stochastic neighborhood embedding (t-SNE) clustering. Clusters derived from the melanoma and skin control samples are shown in blue and red, respectively. The expression of IFN-γ, KI-67, CD45RO, CD69, and PD-1 was used for clustering. (**C**) Histograms of the normalized expression of the parameters used for the clustering in (**B**), melanoma (blue) and skin control (red) clusters. (**D**) Heatmap of the normalized expression of the parameters used in (**B**). (**E**) Topological identity of melanoma CD8 T cell clusters obtained in (**D**). Cluster 6 (green), Cluster 8 (brown). (**F**) Histograms of the normalized expression of the parameters used for the clustering in (**B**), cluster 6 (green) cluster 8 (brown).

## Data Availability

Data will be available upon request to the corresponding author.

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
