# Peer review of "Acral Melanoma Is Infiltrated with cDC1s and Functional Exhausted CD8 T Cells Similar to the Cutaneous Melanoma of Sun-Exposed Skin"

_ijms, 2023, doi:10.3390/ijms24054786_

Round 1

Reviewer 1 Report

The authors of the presented manuscript aim at identification of cCD1s and CD8 T cells infiltrating acral melanoma in comparison with cutaneus melanoma. As authors clearly state, scientific papers aiming at immune cells infiltration of this particular tumour are scarce due to its natural prevalence in non-caucasian population. Most of the melanoma clinical trials are being carried out in countries with a predominantly Caucasian population, and thus, they are more indicative of the response of cutaneous melanoma than of the other subtypes of melanoma. In this manuscript, authors present valuable data predicting comparable sensitivity of acral and cutaneus melanoma to immunotherapy, which represent promising new treatment strategy of acral melanomas.

To improve quality of this paper, please adress the following points:

1. Introduction and discussion can be shortened as some information are mentioned more than once

2. Figure 1A needs magnification/higher resolution

3. Authors should discuss their results with results of other authors showing legitimacy of immunotherapy in acral melanoma treatment (e.g. Li, Jiannong et al. “Single-cell Characterization of the Cellular Landscape of Acral Melanoma Identifies Novel Targets for Immunotherapy.” Clinical cancer research : an official journal of the American Association for Cancer Research vol. 28,10 (2022): 2131-2146. doi:10.1158/1078-0432.CCR-21-3145

Author Response

We thank the reviewer for helping us to improve the quality of the study, and in the next pages, we proceed to give a point-by-point answer to the reviewers´ comments. Changes to the original manuscript are highlighted either with the World´s function tracker of changes, and major changes or new paragraphs are highlighted with yellow. 

Comments and Suggestions for Authors

The authors of the presented manuscript aim at identification of cCD1s and CD8 T cells infiltrating acral melanoma in comparison with cutaneus melanoma. As authors clearly state, scientific papers aiming at immune cells infiltration of this particular tumour are scarce due to its natural prevalence in non-caucasian population. Most of the melanoma clinical trials are being carried out in countries with a predominantly Caucasian population, and thus, they are more indicative of the response of cutaneous melanoma than of the other subtypes of melanoma. In this manuscript, authors present valuable data predicting comparable sensitivity of acral and cutaneus melanoma to immunotherapy, which represent promising new treatment strategy of acral melanomas.

To improve quality of this paper, please adress the following points:

  1. Introduction and discussion can be shortened as some information are mentioned more than once.

Response: We have eliminated some parts of the three first paragraphs of discussion. This information is now in the introduction. However, the introduction section has been extended as suggested by other reviewer.

  1. Figure 1A needs magnification/higher resolution

Response: We have increased the size and resolution of Figure 1A in the modified version of the manuscript.

  1. Authors should discuss their results with results of other authors showing legitimacy of immunotherapy in acral melanoma treatment (e.g. Li, Jiannong et al. “Single-cell Characterization of the Cellular Landscape of Acral Melanoma Identifies Novel Targets for Immunotherapy.” Clinical cancer research : an official journal of the American Association for Cancer Research vol. 28,10 (2022): 2131-2146. doi:10.1158/1078-0432.CCR-21-3145

Response: We also consider this report very relevant and we have previously cited as reference #31. The study is now still mentioned in the introduction, but also in the 3er paragraph of discussion as follows:.

“Some studies support that AM is inherently more immune-silent than NACM. For instance, Li Jiang and colleagues observed by single-cell RNA-sequencing that AM was less infiltrated than NAMC, and the infiltrate presented a more exhausted phenotype [53]. Our data differ from those presented in this study. Although the origin of this discrepancy between studies is unclear, it could be explained by the different transcriptomic and proteomic approaches used”.

Reviewer 2 Report

The manuscript entitled " Acral Melanoma is Infiltrated with cDC1s and Functional Exhausted CD8 T Cells Similar to the Cutaneous Melanoma of Sun-Exposed Skin" presented the possible role of cDC1 and CD8 lymphocytes in the pathogenesis of acral melanoma. The manuscript contains an adequately described research methodology; however, I think several elements should be improved to increase the value of the presented research.

Major concerns regarding this study:

1.     Introduction part: This part of the manuscript is compact but also a little incoherent, thus making it impossible to define the topic/purpose of particular paragraphs clearly. Additionally, the authors assess the immunological aspect by evaluating dendritic cells and CD8 lymphocytes. Therefore, I recommend a concise introduction to the fundamental immune aspects involving that cells in anti-tumor response.  Moreover, authors should provide information about why they chose precisely that one’s subset, especially in the wide spread of different populations of immune cells.

2.     Line 97-105: I would recommend removing these lines from the introduction part because they do not constitute the substantive subject of the discussed issue but only presents the role of The Mexican Institute of Social Security (IMSS) in restoring public health in Mexico.

3.     The characteristics of the study group should also be supplemented with basic information such as the patient's age, sex, length of exposure to UV radiation, etc. The appropriate way will be a supplementary table involving a stratification based on the clinical stage. Mentioned information would undoubtedly raise the level of the presented manuscript.

4.     The authors should provide information about the number of cells used for the flow cytometry assessment.

5.     Line 613 and 617-618: Lack of the clone numbers of mentioned antibodies. Please provide it.

6.     Authors should include a representative gating strategy together with Isotype control based on the flow cytometry assessment in supplementary materials.

7.     Figure 1d. Lack of scale visualization; please provide it.

8.      I would like to recommend including the information on the basis of the algorithm used for establishing the cellular density in ImageJ program (how this density was calculated) instead of a reference to the previous publication.

9.     Figures showing multi-parametric staining (confocal microscopy) should be accompanied by charts with a single stain control as a supplementary figure to exclude potential autofluorescence phenomenon, e.g., Additionally, confocal graphs should be provided in better quality, it’s hard to assess the data presented on its.

10.  In figures 2c-e, the authors should provide an exact marking of the Y axis.  What the “% “does means? Frequency of mononuclear cells or dermis cells? The same remark to Fig3c-e.

11.  The authors mention the expression of PD-1 as one of the examples of the exhaustion of immune cells. This is not a mistake. Nevertheless, it should be remembered that it is also an immune checkpoint expressed on activated T cells, NK cells, and B cells, which plays a crucial role in promoting self-tolerance by downregulating the immune response. The role of immune checkpoints has been widely described in the literature for several years, also in the context of melanoma. Therefore, assessment of the exhausted phenotype of immune cells only based on the expression of PD-1 is not good practice. The assay should also consider additional markers such as LAG3, TIGIT, and KLRG1. Regarding this, I also recommend changing the discussion part based on the theory of exhausted cells. There is a courageous conclusion that requires additional confirmation.

12. Discussion: it is difficult to determine the relevant data corresponding to the obtained results – it is not precise. Please focus on the comparison of your results with the actual literature

Author Response

We thank the reviewer for helping us to improve the quality of the study, and in the next pages, we proceed to give a point-by-point answer to the reviewers´ comments. Changes to the original manuscript are highlighted either with the World´s function tracker of changes, and major changes or new paragraphs are highlighted with yellow. 

Comments and Suggestions for Authors

The manuscript entitled " Acral Melanoma is Infiltrated with cDC1s and Functional Exhausted CD8 T Cells Similar to the Cutaneous Melanoma of Sun-Exposed Skin" presented the possible role of cDC1 and CD8 lymphocytes in the pathogenesis of acral melanoma. The manuscript contains an adequately described research methodology; however, I think several elements should be improved to increase the value of the presented research.

Major concerns regarding this study:

  1. Introduction part: This part of the manuscript is compact but also a little incoherent, thus making it impossible to define the topic/purpose of particular paragraphs clearly. Additionally, the authors assess the immunological aspect by evaluating dendritic cells and CD8 lymphocytes. Therefore, I recommend a concise introduction to the fundamental immune aspects involving that cells in anti-tumor response.  Moreover, authors should provide information about why they chose precisely that one’s subset, especially in the wide spread of different populations of immune cells.

Response: Following your observation, we have extended the information about cDC1s and CD8 T lymphocytes in paragraphs 3er and 4th of the introduction. We hope that we now clearly explain the importance of addressing these immune populations.

  1. Line 97-105: I would recommend removing these lines from the introduction part because they do not constitute the substantive subject of the discussed issue but only presents the role of The Mexican Institute of Social Security (IMSS) in restoring public health in Mexico.

Response: We believe that it is important to describe the source of the patients, since we observed a disproportionate number of acral melanomas compared with Hispanics in the USA and with previous Mexican epidemiological studies. We consider that the explanation for the enrichment of acral melanoma in the study is explained by the population attended at IMSS Oncology Hospital. Because the description of IMSS may be distracting, we have moved it to Material and Methods, Section 4.1 Human Melanoma and Control Samples.

  1. The characteristics of the study group should also be supplemented with basic information such as the patient's age, sex, length of exposure to UV radiation, etc. The appropriate way will be a supplementary table involving a stratification based on the clinical stage. Mentioned information would undoubtedly raise the level of the presented manuscript.

Response: Following your suggestion, we have added Supplementary Table 1 that includes the clinical and demographic information of the patients. Since we are unable to asses the length of exposure to UV radiation, we have included a column in which we describe the exposure as high or low according to whether the site of the primary lesion is on an area of the body exposed or hidden from the sun light.

  1. The authors should provide information about the number of cells used for the flow cytometry a

Response: We have added this information to the 4.7 Flow Cytometry Staining section of Materials and Methods. 100,000 and 245,000 total events were acquired in Skin control and tumor infiltrating lymphocytes, respectively. To analyze memory CD8 T cells, 10,000 events of CD45RO+ CD8+ cells were collected.

  1. Line 613 and 617-618: Lack of the clone numbers of mentioned antibodies. Please provide it.

Response: We have added a Supplementary Table 2 with all the information regarding the antibodies used in this study.

  1. Authors should include a representative gating strategy together with Isotype control based on the flow cytometry assessment in supplementary materials.

Response: The gating strategy and controls are now included in Supplementary Figure 9.

  1. Figure 1d. Lack of scale visualization; please provide it.

Response: A scale bar is now included.

  1. I would like to recommend including the information on the basis of the algorithm used for establishing the cellular density in ImageJ program (how this density was calculated) instead of a reference to the previous publication.

Response: Following your suggestion, we have expanded the description of the method of analysis and how we measured cellular density. This new information is included in the section 4.6 Immunofluorescence Image Analysis of M&M.

“The IF analyses were performed using our Machine Learning method described in [51]. This method involves automatic nuclei segmentation using our model trained in Convolutional Neuronal Network Cellpose. The model subsequently measures the expression of different markers to define the phenotype with Annotater in ImageJ. The results obtained were analyzed using Python scripts. The percentage of positive cells was calculated by dividing the total number of cells positive for a given phenotype by all cells in the field multiplied by 100. Density was measured by dividing the number of cells of interest by the total area of the field. The results shown in the graphs represent the median value of three analyzed areas per sample/patient. Although fibrotic areas and blood vessels can exhibit high autofluorescence, these areas were not included in the experimental assessments and comparisons between AM and NACM, for which only marks with nuclei were analyzed. In the case of the XY coordinate maps and density contour maps, the information obtained with our Machine Learning image analysis method was plotted using the CytoMap Software [95]”.

  1. Figures showing multi-parametric staining (confocal microscopy) should be accompanied by charts with a single stain control as a supplementary figure to exclude potential autofluorescence phenomenon, e.g., Additionally, confocal graphs should be provided in better quality, it’s hard to assess the data presented on its.

Response: The new Supplementary Figures 1 to 8 include all the different staining controls used for every experiment in the study.

Supplementary Figures 1, 2 and 3 show controls in which we added the fluorochrome-coupled secondary antibody without primary antibodies.

Supplementary Figures 5 and 7 show the signals for all the individual staining performed in the multiparametric immunofluorescence experiments of Figures 5 and 6. Supplementary Figures 6 and 8 show the acquired images of every channel after each bleaching cycle in order to illustrate how the signal of the fluorophores is lost.

We also provide images with better quality for optimal appreciation of the results.

  1. In figures 2c-e, the authors should provide an exact marking of the Y axis.  What the “% “does means? Frequency of mononuclear cells or dermis cells? The same remark to Fig3c-e.

Response: In the modified version of the manuscript, we depict in the legend of Figure 2 and 3 that the percentages of cDC1s and CD8 T cells, respectively, were estimated among all nucleated cells present in the selected highly infiltrated areas of the melanoma.

  1. The authors mention the expression of PD-1 as one of the examples of the exhaustion of immune cells. This is not a mistake. Nevertheless, it should be remembered that it is also an immune checkpoint expressed on activated T cells, NK cells, and B cells, which plays a crucial role in promoting self-tolerance by downregulating the immune response. The role of immune checkpoints has been widely described in the literature for several years, also in the context of melanoma. Therefore, assessment of the exhausted phenotype of immune cells only based on the expression of PD-1 is not good practice. The assay should also consider additional markers such as LAG3, TIGIT, and KLRG1. Regarding this, I also recommend changing the discussion part based on the theory of exhausted cells. There is a courageous conclusion that requires additional confirmation.

Response: As we describe in the 6th paragraph of discussion, we agree that PD-1 expression marks early activation of T cells in addition of exhaustion. We believe that PD-1 low to mid expression is more likely found in functional cells, while PD-1 high expression is enriched in terminally non-functional exhausted cells. In agreement, when we evaluated TIM-3 expression, we observed an enrichment of TIM-3 in the PD-1high CD8 T cells (see Supplementary Figure 4). Likewise in the flow cytometry analysis (see Supplementary Figure 9). Here we observed that the cell clusters exclusively present in melanoma samples are positive to expression of PD-1, TIM-3 and CD39. More in-depth characterization of exhaustion and functional markers is needed to accurately assign functional and terminally exhausted CD8 T cells. The main aim of this study was to provide evidence of the presence of cDC1 and CD8 T cells in the stroma of acral melanoma, and that these cells are expressing functional molecules and checkpoint inhibitors, which made them a potential target for immunotherapy. We have also softened the interpretation that the PD-1+ cells are exhausted cells, and we only suggest their functional state based on the expression of both, markers of effector responses and of non-functional exhaustion.

  1. Discussion: it is difficult to determine the relevant data corresponding to the obtained results – it is not precise. Please focus on the comparison of your results with the actual literature

Response: We have narrowed the discussion section. Particularly, we have eliminated the general information found in the first paragraphs to emphasize only the discussion relevant to the data obtained in the study.

Round 2

Reviewer 2 Report

The authors have considered all my major concerns in the revised version of the manuscript. However, there are still a few minor language errors that will probably be included in further editing processes.